# Early Onset Intrauterine Growth Restriction—Data from a Tertiary Care Center in a Middle-Income Country

**DOI:** 10.3390/medicina59010017

**Published:** 2022-12-21

**Authors:** Marina Dinu, Anne Marie Badiu, Andreea Denisa Hodorog, Andreea Florentina Stancioi-Cismaru, Mihaela Gheonea, Razvan Grigoras Capitanescu, Ovidiu Costinel Sirbu, Florentina Tanase, Elena Bernad, Stefania Tudorache

**Affiliations:** 18th Department, Faculty of Medicine, University of Medicine and Pharmacy of Craiova, 200349 Craiova, Romania; 21st Department, Faculty of Medicine, University of Medicine and Pharmacy of Craiova, 200349 Craiova, Romania; 3Obstetrics and Gynecology Department, Mioveni City Hospital, 115400 Mioveni, Romania; 4Obstetrics and Gynecology Department, Emergency County Hospital, 200349 Craiova, Romania; 5Obstetrics and Gynecology Department, Faculty of Medicine, “Victor Babes” University of Medicine and Pharmacy, 300041 Timisoara, Romania; 6Obstetrics and Gynecology Department, “Pius Brinzeu” County Clinical Emergency Hospital, 300723 Timisoara, Romania

**Keywords:** early onset intrauterine growth restriction, ultrasound, uterine artery pulsatility index, tertiary care center, middle-income country, number of prenatal visits

## Abstract

*Background and Objectives:* In this study, we aimed to describe the clinical and ultrasound (US) features and the outcome in a group of patients suspected of or diagnosed with early onset intrauterine growth restriction (IUGR) requiring iatrogenic delivery before 32 weeks, having no structural or genetic fetal anomalies, managed in our unit. A secondary aim was to report the incidence of the condition in the population cared for in our hospital, data on immediate postnatal follow-up in these cases and to highlight the differences required in prenatal and postnatal care. *Materials and Methods*: We used as single criteria for defining the suspicion of early IUGR the sonographic estimation of fetal weight < p10 using the Hadlock 4 technique at any scan performed before 32 weeks’ gestation (WG). We used a cohort of patients having a normal evolution in pregnancy and uneventful vaginal births as controls. Data on pregnancy ultrasound, characteristics and neonatal outcomes were collected and analyzed. We hypothesized that the gestational age (GA) at delivery is related to the severity of the condition. Therefore, we performed a subanalysis in two subgroups, which were divided based on the GA at iatrogenic delivery (between 27+0 WG and 29+6 WG and 30+0–32+0 WG, respectively). *Results:* The prospective cohort study included 36 pregnancies. We had three cases of intrauterine fetal death (8.3%). The incidence was 1.98% in our population. We confirmed that severe cases (very early diagnosed and delivered) were associated with a higher number of prenatal visits and higher uterine arteries (UtA) pulsatility index (PI) centile in the third trimester—TT (compared with the early diagnosed and delivered). In the very early suspected IUGR subgroup, the newborns required significantly more NICU days and total hospitalization days. *Conclusions:* Patients with isolated very early and early IUGR—defined as ultrasound (US) estimation of fetal weight < p10 using the Hadlock 4 technique requiring iatrogenic delivery before 32 weeks’ gestation—require closer care prenatally and postnatally. These patients represent an economical burden for the health system, needing significantly longer hospitalization intervals, GA at birth and UtA PI centiles being related to it.

## 1. Introduction

Severe early-onset intrauterine growth restriction (IUGR) complicates around 0.4% of pregnancies [1,2,3] and is associated with poor and very poor pregnancy outcome due to high morbidity and mortality. This is related primarily to premature iatrogenic delivery both for fetal and for maternal indications [4]. Placental disease is associated with a low volume of uteroplacental blood flow and a spectrum of hypertensive disorders. Thus, these cases are often referred to tertiary centers.

The neonatal intensive care unit (NICU) stay is required in most cases and the long-term neurodevelopmental sequelae are important, affecting more than two-thirds of these babies. Survival rates for extremely early born growth-restricted babies (<28 weeks’ gestation—WG) vary from 7% to 33% [4,5,6]. Neonatal morbidity is gestational age (GA) related [7] and related to the severity of IUGR also [8].

The costs of this population of fetuses/neonates include the cost of increased antenatal surveillance (with or without hospitalization days), caesarean delivery, NICU care, routine post-NICU follow-up, and specialized neurodevelopmental assessments and interventions. Such costs represent an important economic burden, especially in developing and middle-income countries. Safe pregnancy prolongation implies a higher number of prenatal consultations [9].

Doppler waveform analysis in pregnancies complicated by IUGR helps in confirming/ruling out the compromise of uteroplacental circulation and placental hypoperfusion. Currently, there are no specific evidence-based therapies for placental insufficiency and for early-onset severe IUGR. Bed rest and hospital admission for surveillance are not scientifically supported by randomized controlled trials. Many management strategies were proposed and studied, including medical interventions, such as Sildenafil citrate [2,3].

IUGR remained the second leading cause of perinatal mortality following prematurity [10]. It has significant consequences on neonatal, childhood and adult morbidity [11]. Currently, there have been scarce reports regarding early-onset IUGR in populations in Romania. This study aimed to assess the prevalence at birth of early-onset IUGR requiring preterm birth before 32 WG in a tertiary center and its associated factors. The end-target is to follow up long-term this population of newborns.

## 2. Materials and Methods

We performed a nested cohort prospective study. It was designed and conducted in the Prenatal Diagnosis Unit of the Emergency County Hospital of Craiova, which is a tertiary referral university-affiliated Hospital in the south-west region of Romania.

The study included singleton pregnancies having an estimated fetal weight less than the 10th percentile (<p10) at any scan between 22 and 32 WG and no known structural or genetic abnormality. We used the Hadlock 4 technique [12] for the US estimation of fetal weight (EFW). The cases falling under p10 (thus defined as suspected of having early IUGR) were enrolled consecutively between 22- and 31+6 WG.

The study was carried out over a period of three years (1 September 2019–1 September 2022). We report data on 36 pregnancies with prenatal and postnatal care provided in our hospital (complete follow-up, delivery, and postnatal care).

We used a poststudy selected control group. In this group, we included 56 cases of normal pregnancies. The cases were retrospectively selected, consecutively, from the population completely followed up and delivered in our hospital following the study beginning date, September the 1st 2019: healthy mothers having singleton normal fetuses (in terms of structure and growth curve) with pregnancies resulting in normal vaginal term uncomplicated births.

Even if included in a low-risk pregnancy group at registration, all women having prenatal care in our unit are offered and scheduled for the end of the first trimester (detailed anomaly and “genetic” scan [13,14]), for a second trimester (structural survey—anomaly scan) and a third trimester (well-being) US scan. If the prenatal exams (dating and FT anomaly and genetic scan) lead to completely normal data, for the ST scan, the GA offered is 20–23 weeks, and for the TT, it is 29–33 weeks.

In the study group, we included cases requiring hospitalization and/or followed up as outpatients.

We included exclusively pregnancies with a known gestational age (GA) confirmed by US during the first trimester (before 13 weeks 6 days). Patients with fetal structural/chromosomal anomalies, uncertain gestational age and/or unavailable complete data were excluded from the analysis.

We used for all cases a Voluson E10 (GE Medical Systems, Chicago, IL, USA) ultrasound machine equipped with a 4–8 MHz curvilinear transducer. When using color Doppler, the mechanical and thermal indices were kept as low as possible (ALARA principle) [15], and safety guidelines were followed [16].

All scans were performed by the author (obstetrician sonographer M.D.) and—in selected cases—repeated by a senior consultant (S.T.). The study protocol was approved by the university ethics committee, and informed consent was obtained from all participants prior to enrolment.

Internal policy adjusted to current guidelines [17,18] was applied regarding the administration of antenatal steroids for fetal lung maturity and magnesium sulfate for fetal neuroprotection.

We used for the uterine arteries (UtA) [14,19], umbilical artery (UmbA) [14,20], middle cerebral artery (MCA) [14,21] and ductus venosus (DV) Doppler [14,22] assessment the technique previously recommended. We also calculated the cerebroplacental ratio (CPR) as previously described [14,23].

We chose to report separately the Doppler indices for each uterine artery instead of reporting the median of both. The observer diagnosed lateral placenta if more than half of the placenta was seen on US on one side of the uterine cavity only. The corresponding (right or left) uterine artery was named “placental”. The other one was named “non-placental” uterine artery on the US form.

The timing of delivery was customized based on the gestational age, the severity of the disease—depending on the results of fetal surveillance, the parents’ decisions, and a team of senior consultants, including neonatologists. If the internal policy was not changed by the attending physician (increasing or decreasing the intensity of prenatal care and the frequency of medical visits) or on parental desire (e.g., transfer to another unit or fetal abandonment), we proceeded as follows:
○In prestage I (defined as EFW between 10th centile and the 3rd centile), we used weekly US monitoring regardless of the GA—amniotic fluid volume assessment (using the deepest vertical pocket technique—DVP [24]), fetal biophysical profile (BPP) [25] and Doppler interrogation at the two fetal sites (UmbA and MCA), the CPR, both UtA and pulsatility index (PI); in this stage, we used US for EFW every two weeks. If the BPP and Dopplers were normal, expectance was proposed until 32 WG, and the case was discarded from the study. If the BPP was abnormal, the case was followed up daily. If there was a persistent abnormal BPP (below 5, two days consecutively), we performed elective C-section before 32 WG, regardless of the Doppler results.○In stage I by Figueras [26] (EFW < 3rd centile or CPR < 5th centile or any UtA PI > 95th centile), we offered the same weekly monitoring protocol and the same management. If the BPP was normal, we monitored until we registered the case as advancing toward stage II or until progressing over 32 WG. If the BPP was abnormal, the case was followed up daily. If the BPP was persistently abnormal (below 5, two days consecutively), we added the DV assessment, and we performed an elective C-section before 32 WG regardless of the Doppler results.○In stage II by Figueras [26]—defined as UmbA absent end-diastolic velocity (AEDV)—we offered hospitalization. If the parents declined admittance, we re-examined twice a week. Inpatients were also offered twice-weekly additional cardiotocography (CTG) and DV assessment daily. In this stage, we performed an elective C-section before 32 WG in all cases.○In stage III by Figueras [26]—defined as UmbA reversed end-diastolic velocity (REDV), we monitored cases by US daily. In surviving fetuses, we offered delivery by cesarean section before 30 weeks based on the DV assessment.○In stage IV by Figueras [26]—defined as reversed flow ductus venosus (DV), we offered immediate delivery after 27 weeks by caesarean section to all couples. Benefits and expectations were extensively explained to the parents in these cases.

Demographic data and maternal baseline characteristics, as well as data regarding the course of pregnancy and newborn outcomes, were collected prospectively for the study group and retrospectively (using the institution’s computerized database containing the patient’s antenatal/intra/postnatal records) for the control group.

In the study group, all US data were collected more than once, according to the study design. The statistical analysis was performed on the values at the beginning of the specific trimester (the second or the third). Therefore, the processed values were the ones obtained at 22 weeks’ or at the first prenatal visit in our unit—in all cases enrolled in the ST. The data entering in the final analysis were obtained at 28 weeks’ gestation (in all cases already enrolled) or at the first prenatal visit (at enrollment) in cases enrolled or referred to our unit in the TT.

We collected maternal data and demographics, pregnancy complications, prenatal care, US prenatal features, and postnatal data in newborns.

We perform routine screening for gestational diabetes mellitus (GDM). We use the one-step approach: oral glucose tolerance test (OGTT) at 24–28 WG (without prior plasma or serum glucose screening). We use a 75 g glucose load, and the glucose threshold values are: for fasting—95 mg/dL, at 1 h—180 mg/dL and at 2 h later—155 mg/dL. We classified the patient as positive in this study if two or more of the venous plasma concentrations were met or exceeded [27].

Maternal blood pressure was measured automatically with a calibrated OMRON M6 Confort device, according to standard procedure. Blood pressure was measured in one arm (right or left) without distinction, while women were seated and after a 5 min rest. We defined gestational hypertension as a systolic blood pressure of 140 mm Hg or more or a diastolic blood pressure of 90 mm Hg or more, or both, on two occasions at least 4 h apart after 20 weeks of gestation in a woman with a previously normal blood pressure [28]. We defined preeclampsia as a systolic blood pressure of 140 mm Hg or more or a diastolic blood pressure of 90 mm Hg or more with 300 mg or more of proteinuria. In the absence of proteinuria, new-onset hypertension was determined with the new onset of any of the following: thrombocytopenia: less than 100.000/mm^3^; renal insufficiency; impaired liver function: elevated liver transaminases to twice normal concentration; pulmonary edema; new-onset headache [28]. We defined HELLP syndrome as hemolysis, elevated liver enzymes and low platelet count [29].

We tested for hereditary thrombophilia and defined positive cases if Factor V Leiden homozygote mutation, antithrombin deficiency or protein C or protein S deficiency were found [30].

We performed C-section in all cases, either elective or in emergency circumstances.

We collected data on the newborns during the postpartum hospitalization.

We defined neonatal resuscitation as the set of interventions at the time of birth to support the establishment of breathing and circulation [31]. Respiratory distress was diagnosed if the newborn presented apnea, cyanosis, grunting, inspiratory stridor, nasal flaring, poor feeding, and tachypnoea (more than 60 breaths per minute), retractions in the intercostal, subcostal, or supracostal spaces and if the newborn received surfactant in the therapeutic scheme. Bronchopulmonary dysplasia was diagnosed if fibrotic opacities and cystic changes on the chest imaging X-ray (and on the computed tomography—CT scan) were found. The systemic blood pressure was measured noninvasively in all cases (by means of oscillometric technique, using appropriately sized cuffs). A specific case was reported as positive for hypotension if the abnormal values of systemic blood pressure were documented in the newborn’s file and corrected by volume expansion, inotropes and corticosteroids. Persistent ductus arteriosus (PDA) was suspected on heart murmur and diagnosed by means of postnatal echocardiography. All cases were offered serial transfontanellar ultrasound (on days 3, 7, 14 and at discharge). All newborns benefited from additional heat (warmers and/or incubators). The immature gastro-intestinal (GI) system diagnosis was achieved after excluding other conditions, in babies having feeding intolerances: vomiting, stomach bile, or both; abdominal distension, reduced or absent bowel sounds and reduced or absent stool. All cases received empirical antibiotic treatment. Cases with clinical symptoms and/or with abnormal results on laboratory tests (abnormal white blood cell count, acidosis, hyperglycemia, lethargy, diminished responsiveness, fever, abnormal breathing, and circulatory disorders) were classified as neonatal infection.

We report exclusively data on cases requiring delivery between 27 and 32 completed weeks. By the internal unit’s policy, in cases of severely early restricted fetuses needing iatrogenic delivery before 27 weeks, the parents are repeatedly counselled in multidisciplinary teams, and in utero transfer to superior centers is offered. Cases not requiring ending the pregnancy before 32 completed weeks (those continuing the pregnancy later than 32 weeks) were excluded from the analysis.

### Statistical Analysis

Statistical analysis was performed using Minitab 17 Statistical Software. The distributions of the continuous variables were tested for normal values by using the Anderson–Darling test. Data with a normal distribution were presented as a mean value ± standard deviation (SD); the data that did not have a normal distribution were presented as a median and interquartile rate (IQR). To determine the statistical significance of the differences between the two groups for non-normal data, we used the Mann–Whitney test, comparing the medians (*p* value < 0.05), and for categorical data, we used the Chi-Square Test for Association (*p* value < 0.05).

## 3. Results

We performed the observational study during a three-year interval, and we summarized the workflow in Figure 1.

We had 30.5% self-presented cases and 69.4% referrals for sonography in our case series. Most cases (83.3%) had fetal indications for C-section, and the remaining ones had combined indications (fetal and maternal).

The general characteristics of pregnant women in the study are presented in Table 1. Cases were significantly more likely to be smokers or ex-smokers than controls (*p* < 0.01) and tended to be older (*p* = 0.053).

Among the 36 reported cases, we found in 30 cases exclusively laterally located placentas. In the remaining six cases, with the placenta located rather centrally, the operator decided on subjective criteria the assignment of the uterine arteries.

Both uterine arteries (placental and non-placental) assessed by means of spectral Doppler in the ST and in the TT were abnormal in all cases in the study group.

CPR percentiles were abnormal in the TT in the study group.

The mean gestational age at delivery was 30.7 (27–32) weeks in the study group and 39 weeks (37–41) of gestation in the control group.

In the study group, we had three cases of intrauterine fetal death (incidence 8.3%).

The numbers of prenatal medical follow-up visits (the total number and the third trimester number) in pregnant women included in the study are listed and compared (Table 2).

The newborn data in the study group is summarized below (Table 3).

As expected, the number of postnatal hospitalization days was significantly higher in the suspected early IUGR group vs. the control group. The Apgar score and the number of NICU days are expressed as median. Resuscitation measures were required at birth in almost half of the population. During hospitalization, all newborns presented one or more episodes of transient apnea. In very few cases, hypotension occurred. Persistent ductus arteriosus (PDA) was diagnosed frequently and was treated with anti-inflammatory non-steroid drugs, fluid restriction and/or diuretic drugs. No case required surgical treatment for PDA. All cases of intraventricular hemorrhage were mild. We had no case of large brain bleeding, which was expected to induce permanent brain injury. No newborn developed clinical signs of hypothermia. The single case of necrotizing enterocolitis received surgical treatment in the third day of life. All neonates in the study group had various degree of anemia, and all received blood products or transfusions. All neonates developed jaundice, but most of them had minor forms. Transient hypoglycemia was present in almost half of the cases immediately after birth. We had no case of severe persistent hypoglycemia. Despite the routine empirical antibiotic treatment, we had nine severe cases of neonatal infection. One only case had an early-onset form, while the remaining eight cases were diagnosed with late onset infection.

To describe better the severity and the continuum of the disease in the study group, we chose to perform a subanalysis and to compare the antenatal and the postnatal data in 12 pregnancies with very early IUGR (requiring iatrogenic delivery between 27 and 29.6 weeks of gestation) and 24 pregnancies in which the delivery was delayed until 30–32 weeks of gestation.

The ultrasound data regarding the UtA centile in the TT in these two subgroups is graphically represented and compared (Figure 2). In the extremely early suspected IUGR group, we found a higher median than in the early suspected IUGR. The boxplot of umbilical artery percentile revealed that in the very early suspected IUGR group, we found a higher median than in the early IUGR group. Based on the Kruskal–Wallis test, the differences between medians are statistically significant (*p* < 0.01).

Total prenatal visits in pregnancy and of TT visits revealed an increased number of medical visits in the very early suspected IUGR subgroup (Figure 3). Based on the Kruskal–Wallis test, the differences between medians are statistically significant (*p* < 0.01).

In the very early suspected IUGR subgroup, we had higher median values for NICU and total hospitalization days (21.5 days and 49 days, respectively) compared to the early suspected IUGR subgroup (Figure 4). Based on the Kruskal–Wallis test, the differences between medians are statistically significant (*p* < 0.01).

There were statistically significant differences between all US parameters in the very early suspected IUGR subgroup compared with the early suspected IUGR subgroup.

## 4. Discussion

IUGR reflects an abnormal adaptive fetal growth in a deleterious environment. Among all the modalities we have available to assess a fetus—we still do not know how each of them (EFW, Doppler velocities, BPP score), isolated or in combination, will perform in IUGR diagnosis and/or in deciding the time of the delivery [32]. Our data may be used in forthcoming logistic and linear repression analyses needed to prove the independent predictors for long-term outcome.

Our study targeted a very limited population of IUGR fetuses requiring early iatrogenic birth before 32 weeks. We confirmed the known association with hypertensive disorders in pregnancy [33], which is present in 70% in this case series. We searched for associations with non-modifiable (hereditary thrombophilia [34]) and modifiable risk factors (smoking) [35,36].

Early-onset IUGR has significant risks for major and minor neonatal morbidity [37]. We confirmed that the neonatal care is influenced by the severity of prematurity. The NICU days were significantly higher in the very early suspected/diagnosed and delivered group.

We also confirmed the recent reported high overall survival rates in IUGR suspected before 32 WG [37]. We registered three fetal deaths in the study group, having an overall in utero mortality of 8.3%. Among them, two fetuses did not benefit from medical management (fetal abandonment was decided by the parents). One fetus had the EFW < p3 at 26 WG, and the fetal demise occurred between two consecutive follow-up visits, at 29 WG. We registered one only neonatal death in our cohort. Previous reports [38] showed 6% mortality in the IUGR group and 24% severe morbidity.

We have no treatment for IUGR. The sole intervention having some treatment-like effect is the early iatrogenic termination of pregnancy [39]. Yet, the antenatal detection of inadequate fetal growth leads to increased surveillance and reduces the risk of fetal death [40]. According to some results, the prenatal diagnosis may also improve perinatal morbidity [41], although the scientific proof of this statement is still debated [42]. In our case series, the intensity of prenatal care was amplified in early and very early suspicion of IUGR. The total number of prenatal visits in pregnancy and the total number of TT visits was significantly higher in the very early suspected IUGR subgroup. Both sets of figures are much higher than the number recommended by the current guidelines in low-risk pregnancies [43,44,45]. Defensive medicine may play a role in these results, but it cannot be weighted from these data.

We did not report the CPR centiles to describe this population of fetuses, although CPR proved to be superior to the UmbA Doppler assessment in the prediction of adverse perinatal outcome [46,47] and in the prediction of long-term developmental problems [48]. Currently, there are no clinical trials investigating the effectiveness of the CPR in guiding clinical management in IUGR, and it is still unclear to which subgroup of pregnant women this applies best. In our case series, this parameter was abnormal in all cases in the TT. Regarding the maternal interface, we confirmed [49] that all searched parameters were abnormal in both trimesters.

A cost-analysis to follow our report may be appropriate due to the high number of US scans, NICU and total hospitalization days needed in this high-risk population. Our results have the potential to help local authorities in the healthcare system plan an adequate strategy (primary care, medical education, audit, merged databases in university centers, funding), adjusted for emergency state hospitals. Results may lead to appropriate centralization to improve the neonatal outcome.

As limitations, we provide no long-term data on the neonates included in the study.

In this study, the US expertise of the primary referring doctors was not investigated. In Romania, obstetricians ultrasonographers are the main healthcare provider responsible for the assessment of fetal growth in low-risk pregnancies, and TT scan is optional. The primary doctors’ skills are important for identifying impaired fetal growth and referring the mother to a customized prenatal care. Unless placed in an at-risk category, the pregnancy will not be monitored appropriately.

In IUGR, an impressive amount of recent research was published. Definitions of IUGR and significant predictors varied largely throughout the last decades: AC < p10 and UmbA PI > p95 [38] (consensus amongst 20 European experts in perinatology), AC < p10 or EFW < p5 and UmbA PI > p95 [50] (very wide GA considered). We defined “suspected IUGR” as the US EFW less than the 10th centile prior to delivery. We are aware that most researchers define IUGR as two components associated: small size and functional evidence of placental impairment (abnormal Dopplers). We acknowledge the risk of including a certain proportion of small for gestational age healthy fetuses (small sized and having normal results on Doppler interrogation). Yet, we may assume that this population was very limited, since we excluded all pregnancies continuing at 32 WA, and we had no case of pregnancy with normal fetal BPP and normal velocities iatrogenically interrupted, regardless the pEFW.

It has been shown that multivariable Integrative models (using additionally maternal characteristics and maternal biochemical markers) offer only modest improvement in the detection of IUGR when compared with screening based on EFW centile alone [51].

We did not assess the data immediate before delivery. This might have an impact on results due to the dynamic of US parameters [52].

We centered the study in a state hospital, having issues in subsidizing some of the already known strategies to improve the IUGR detection in the antenatal period: FT maternal serum placental growth factor (PlGF) and soluble forms-like tyrosine kinase-1 (sFLT) [51,53].

In our view, this report has also some advantages: we provided data on the characteristics of the mother, US features and the type of prenatal surveillance, covering three years, in a single tertiary center in an upper middle-income country.

Data obtained by a single operator using a single US equipment and a standardized technique for Doppler interrogation assured homogeneity in this study. This has the potential to lead to consistent results, given the considerable methodological heterogeneity in studies reporting reference ranges for UmbA and MCA Doppler indices and CPR. Using different references has important implications for clinical practice [51]

We did not use CTG and short-term variation of fetal heart rate in this population of fetuses suspected of early-onset IUGR, as scientific proof for its benefits is still missing [53].

We had the opportunity to use the hospital’s electronic records, which improved the retrospective collection of data in the control group.

In our view, the contextual factors should be considered. Our study interval overlapped the pandemics, and this heavily impacted the internal policy of the unit, the continuity of care and the rate of admittance. This resulted most probably in biasing the population selection, budgets, staffing, workload, safety, the practice climate, and the management decisions. On the other hands, the attempt to use the same guidelines in different countries without local validation may be difficult, given the differences in the prevalence of adverse pregnancy outcomes in different settings. The prevalence and the severity of a disease influences the diagnostic performance; thus, context-specific guidance is necessary. Given the local reporting gaps about the predictive ability of antenatal Doppler for adverse pregnancy outcomes and for the pregnancy care costs, our data on a very high-risk fetal population may prove informative.

## 5. Conclusions

Fetuses with isolated very early and early fetal growth restriction—defined as ultrasound estimation of fetal weight < p10 using the Hadlock 4 technique and requiring delivery before 32 weeks’ gestation—are likely to be scanned more frequently, and newborns have longer hospitalizations. GA at iatrogenic birth and UtA PI centiles are related to the latter. In developing and middle-income countries, cost-analysis studies should be developed in the future due to the high number of prenatal visits, scans performed by experts, NICU and total hospitalization days. This would help local authorities in the healthcare system plan an adequate strategy (primary care, medical education, audit, centralization and funding) to improve outcome in these cases.

## Figures and Tables

**Figure 1 medicina-59-00017-f001:**
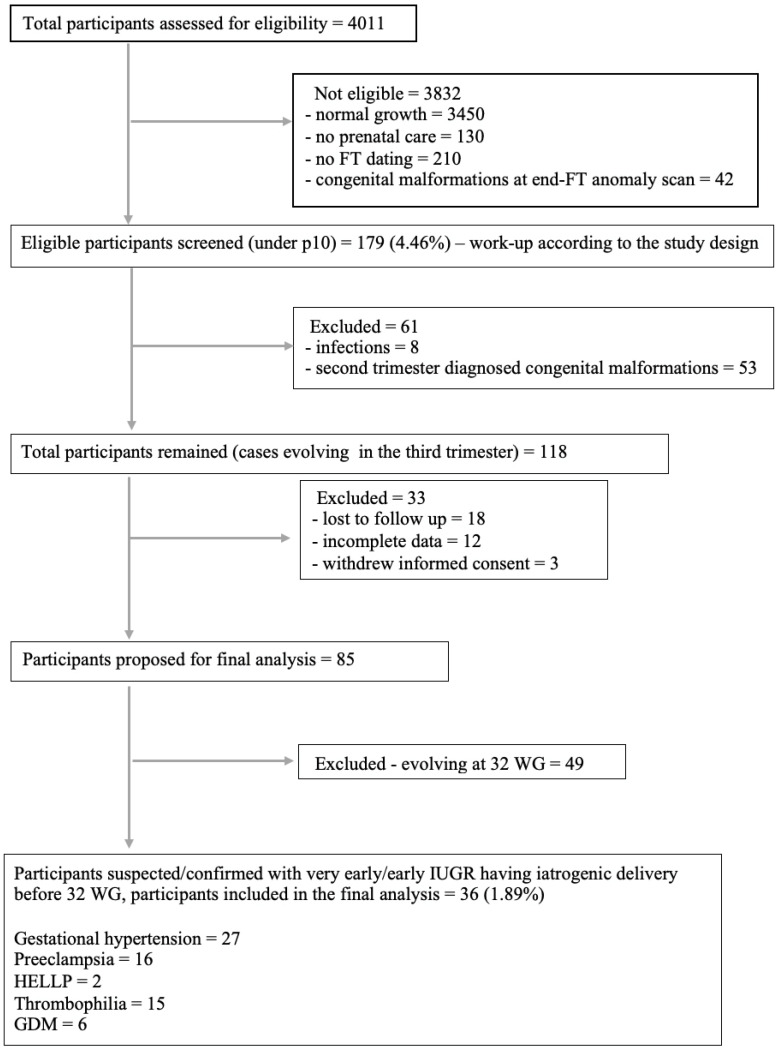
Flow chart diagram.

**Figure 2 medicina-59-00017-f002:**
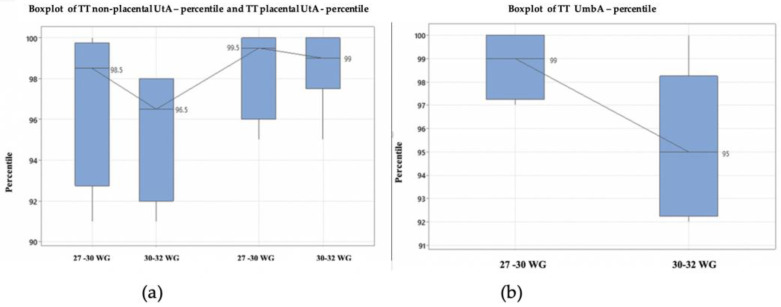
Doppler indices compared in the two subgroups: very early IUGR (requiring iatrogenic delivery between 27 and 29.6 weeks of gestation) and pregnancies which allowed continuing the pregnancy until 30–32 weeks of gestation. (**a**) The boxplot of placental and non-placental uterine artery assessed in the TT; (**b**) The boxplot of umbilical artery percentile. Abbreviations: TT third trimester, UtA uterine artery, WG weeks’ gestation, Umb A umbilical artery.

**Figure 3 medicina-59-00017-f003:**
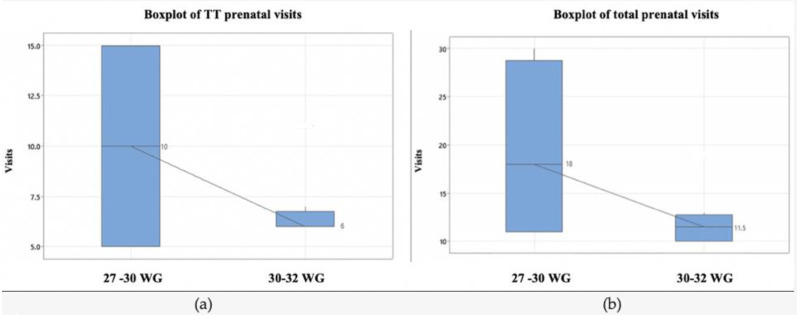
(**a**) The boxplot of TT prenatal visits; (**b**) The boxplot of total number of scheduled medical appointments. Abbreviations: WG weeks’ gestation, TT—third trimester.

**Figure 4 medicina-59-00017-f004:**
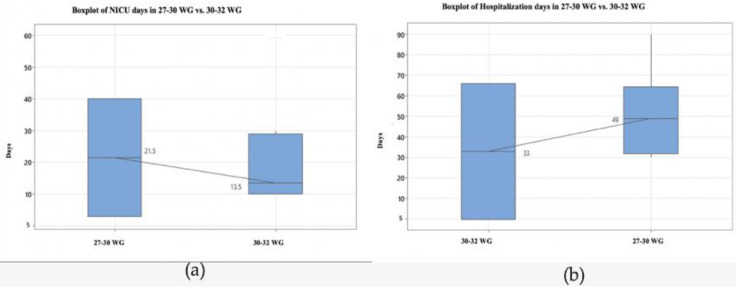
(**a**) The boxplot of NICU days in very early suspected IUGR subgroup vs. in the early suspected IUGR subgroup; (**b**) The boxplot of hospitalization days in the very early suspected IUGR subgroup vs. in early suspected IUGR subgroup. Abbreviations: GA gestational age, NICU neonatal Intensive Care Unit, WG weeks’ gestation.

**Table 1 medicina-59-00017-t001:** Demographic maternal characteristics in pregnancies with suspected early IUGR and control group.

Characteristics	susp Early IUGR	Control	*p*
Smoking/former smoker	66.67%	21.4%	<0.01
Age	29.17 (19–37)	27.17 (18–35)	0.053
BMI	24.5 (19–27)	27.0 (17–31)	0.374

Abbreviations: IUGR intrauterine growth restriction, BMI body mass index. Age and BMI are expressed as median.

**Table 2 medicina-59-00017-t002:** Number of prenatal visits in pregnancy and in the third trimester in the study IUGR group versus the control group.

Variable	susp Early IUGR	Control	*p*
Nr of prenatal visits	11.5 (10–30)	5 (5–6)	<0.01
Nr of prenatal visits in the TT	6 (5–15)	2 (1–3)	<0.01

Abbreviations: TT the third trimester, IUGR intrauterine growth restriction, Nr number.

**Table 3 medicina-59-00017-t003:** Newborn data in pregnancies complicated with early IUGR.

Characteristic/Complications	susp Early IUGR	Controls	*p*
Hospitalization days	36 (22–90)	3.8 (2–5)	<0.01
Apgar Score	5.5 (1–8)		
Resuscitation	17 (47.2%)		
Birth percentile	1% (1–10%)		
NICU days	10.5 (0–60)		
Respiratory Distress Syndrome	20 (55.5%)		
Bronchopulmonary Dysplasia	1 (2.7%)		
Transient Apnea	36 (100%)		
Hypotension	4 (11.1%)		
PDA	15 (41.6%)		
IVH	6 (16.6%)		
PVL	2 (5.5%)		
Hypothermia	0		
Immature GI System	32 (88.8%)		
NEC	1 (2.7%)		
Anemia	36 (100%)		
Jaundice	10 (27.7%)		
Transient Hypoglycaemia	16 (44.4%)		
Infection	9 (25%)		

Abbreviations: NICU Neonatal intensive care unit, PDA Persistent ductus arteriosus, IVH intraventricular hemorrhage, PVL periventricular leukomalacia, GI gastro-intestinal, NEC necrotizing enterocolitis.

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
