# Peer review of "Early Onset Intrauterine Growth Restriction—Data from a Tertiary Care Center in a Middle-Income Country"

_medicina, 2022, doi:10.3390/medicina59010017_

Round 1
Reviewer 1 Report
The Study titled “Early onset intrauterine growth restriction ultrasound – _data 2 from a tertiary care center in a developing country” aimed to define the most significant risk factors and the outcome in a group of patients suspected of or diagnosed with early onset intrauterine growth restriction ending pregnancy before 32 weeks.
The work is written in a structured and understandable manner. There are only minor English corrections to be made.
The method and execution of the work are clearly described.
The discussion is extensive and illuminates all aspects of early growth retardation and accordingly the literature is comprehensive.
The limitations of this study are also described in detail.
Figures and tables are clear and layered.
The only surprising thing about this work is the outcome of both groups, i.e. the study group and the control group. The control group, who gave birth on their (around) estimated due date, did not find any GDM, hypertension, pre-eclampsia, etc. in any of the patients, which is not entirely understandable. was the control group too small? Was the data collection or anamnesis not carried out properly? The fact that the control group showed no pregnancy complications at all (even if only occasionally) is not understandable
This point should be discussed further in the discussion.
Author Response
The Study titled “Early onset intrauterine growth restriction ultrasound – _data 2 from a tertiary care center in a developing country” aimed to define the most significant risk factors and the outcome in a group of patients suspected of or diagnosed with early onset intrauterine growth restriction ending pregnancy before 32 weeks.
The work is written in a structured and understandable manner. There are only minor English corrections to be made.
The method and execution of the work are clearly described.
The discussion is extensive and illuminates all aspects of early growth retardation and accordingly the literature is comprehensive.
The limitations of this study are also described in detail.
Figures and tables are clear and layered.
The only surprising thing about this work is the outcome of both groups, i.e. the study group and the control group. The control group, who gave birth on their (around) estimated due date, did not find any GDM, hypertension, pre-eclampsia, etc. in any of the patients, which is not entirely understandable. was the control group too small? Was the data collection or anamnesis not carried out properly? The fact that the control group showed no pregnancy complications at all (even if only occasionally) is not understandable
This point should be discussed further in the discussion.
We respectfully thank to the reviewer for the kind words, for the time invested to review our manuscript, and for considering our team’s work and results reported.
We thank for the opportunity to clarify the issue raised.
We retrospectively selected a completely normal evolving group of patients and used these cases as controls. We described this group in the method section. We mentioned in page 2, lines 95-99: “We used a poststudy selected control group. In this group we included 56 cases normal pregnancies. The cases were retrospectively selected from the large population completely followed-up and delivered in our hospital: healthy mothers having singleton normal fetuses (in terms of structure and growth curve), with pregnancies resulting in normal vaginal term uncomplicated births”.
The normal cases were selected after the study was completed, sequentially. The first patient was selected at the study beginning (calendar’s date 1st Sept 2019). The last one – after four months. We had (as shown in the flow chart diagram) over 4000 deliveries in our unit in the study period. We suspect that we needed such a long period to be able to include in this control group the cases (56), due to the high incidence of C-section deliveries in our unit (almost 66%).
We do not deny the potential for bias in how they were selected, but we sincerely hope that it is limited: we selected consecutively, all cases that met the inclusion criteria (page 3, lines 102-106):
- completely followed-up and delivered in our hospital,
- healthy mothers,
- singleton pregnancy,
- normal fetus (in terms of structure and growth curve),
- normal (vaginal) uncomplicated birth,
- term birth.
We added information regarding the control group in the manuscript.
We are ready to agree that choosing a completely normal (mothers, fetuses and normal mode of delivery) cohort as controls may be seen as needless.
Yet, this was the study design we choose, in order to highlight the practical consequences of this difficult to predict, difficult to diagnose and difficult to treat condition – the early-onset IUGR.
Kind regards
Reviewer 2 Report
Medicina-2022485 review
I would like to commend the authors for their efforts to improve our understanding of early-onset fetal growth restriction. Unfortunately, the paper in its current form has a number of issues that I feel need to be addressed. I hope you will take my comments and suggestions in the spirit in which they are intended – to make the most of all the hard work you have done.
Summary
This paper provides descriptive data on the outcomes of a case series of early-onset FGR pregnancies requiring delivery before 32 weeks. Such data is important for patient counselling and optimising timing of delivery but varies between healthcare settings. The article also includes a case-control element of questionable value.
Major revisions
1) Study Design
I don’t get a clear sense of what you are trying to achieve with this study and the results and analyses you report don’t seem to have a clear value. To my mind your paper combines a descriptive report of outcomes from early-onset FGR and a case-control study. However, the statistical comparisons you make between your cases and controls don’t appear to answer any questions.
a. Controls
You report the differences in Doppler velocimetry between cases and controls, but your cases were partially selected on the basis of Dopplers while your controls were selected not to have abnormal Dopplers, so they will by definition be different. Similarly, Table 1 reports co-morbidity for cases and controls, but you selected your controls to have no complications. You conclude from this that these co-morbidities are ‘risk factors’ but you cannot draw that conclusion from the study as designed.
Was the aim of the control group to provide a comparison when it comes to antenatal visits and newborn data? There may be some value in this for antenatal visits but has limited value for the newborn data.
If you are going to have a control group I would suggest reconsidering the group you have selected and providing more details about the selection. Why did you limit this to women with no complications having vaginal deliveries? Is this the most meaningful comparison? Were they selected at random or sequentially? Is there potential for bias in how they were selected? Do all women have second and third trimester ultrasound scans in your unit or were your controls selected because they had had scans? If so why were they having scans?
b. Cases
Was there a particular reason you limited your reported outcomes to those delivered before 32 weeks? When seeing these women during pregnancy we don’t necessarily know how things will progress and what will happen to the Dopplers, so providing information about all of these pregnancies could be helpful.
I think the value in your study comes from providing early-onset FGR outcome data from a middle-income country (the World Bank classify Romania as middle-income rather than developing). To this end it would be good to have more data about neonatal morbidity, even just short-term.
You compare Doppler measurements between those delivering 27-30 weeks and those delivering 30-32 weeks, but you also say third trimester scans were done at 28 weeks (or as soon as possible after enrolment). Does this mean some of the 27-30 week group don’t have an included measurement because they had delivered before 28 weeks? Where the measurements done at comparable gestations or were a lot of the 30-32 week group only enrolled at later gestations?
Minor revisions
· Line 139: Did you include pregnancies with an EFW <10th and umbilical artery PI >95th in Stage 1? I know Francesc Figueras doesn’t include umbilical artery PI in his flow chart but he does include it in the table above.
· Figures 1-4: I’m not sure these are really necessary and your AEDV picture is not the clearest as there’s some background noise around the baseline.
· Lines 179-194: you don’t need to report this level of detail in your methods as it will become evident form your results what data you collected.
· Line 196: I don’t find your definition of GDM very helpful in this setting. The ADA paper you cite in turn cites the 4th International Workshop-conference on GDM from 1997 which in turn cites a number of other references and uses the phrase ‘varying degrees’ rather than ‘any degree’. I would argue all pregnant women are slightly more insulin resistant than they were before pregnancy because of the HPL. How did you test for it in your hospital? What cut-offs did you use?
· Line 211: do you mean you tested for thrombophilias?
· Line 225: I assume you mean p<0.05
· Figure 5: I would say that assessing for eligibility and screening are the same thing. You might need to come up with different ways of talking about how and why you excluded people at different stages or put all the exclusions together.
· Line 259: ‘significand’
· Figure 6: how are you defining placental and non-placental? It isn’t completely clear what you’re comparing in these figures but the lines between the medians are not appropriate, as this would be for changes in one group over time. I prefer box plots that are overlayed with the individual data points, as they give a clearer sense of the spread of the data.
Author Response
Medicina-2022485 review
I would like to commend the authors for their efforts to improve our understanding of early-onset fetal growth restriction. Unfortunately, the paper in its current form has a number of issues that I feel need to be addressed. I hope you will take my comments and suggestions in the spirit in which they are intended – to make the most of all the hard work you have done.
Summary
This paper provides descriptive data on the outcomes of a case series of early-onset FGR pregnancies requiring delivery before 32 weeks. Such data is important for patient counselling and optimising timing of delivery but varies between healthcare settings. The article also includes a case-control element of questionable value.
We respectfully thank to the reviewer for the kind words, for the time invested to review our manuscript and for his/her effort to improve the teams’ reported results.
Accumulating evidence demonstrate that being small at birth has future implications for the quality of health during adulthood. It is our aim to follow this population of fetuses on a long term, and to report data on their childhood and adult life.
There is indeed a wide variability in the management of IUGR, in terms of monitoring and recommended gestational age at delivery, mainly in early-onset forms. Clinical practice variability is a marker of poor quality of care. Comparison among studies is difficult and there are no clear recommendations in the literature. Thus, iatrogenic delivery is offered at a wide spectrum of gestational age in early restricted fetuses.
We realize that selecting (in the poststudy period) a completely normal cohort and using these cases as controls does not have a critical importance in answering many questions. Yet, the report has a rather descriptive purpose. The study is committed to share data on a specific setting.
We omitted in the revised form all ultrasound data referring to the control group.
We decided to keep comparing (the study group versus the control group) the following parameters: the maternal body mass index, the maternal age, the habit of smoking, the number of prenatal visits (total and the third trimester) and the hospitalization days in neonates.
The maternal age, the body mass index reached no statistically significant differences. The single risk factor highlighted by the statistical analysis was smoking (former smokers included). This result support the hypothesis that this risk factor may be important in the maldevelopment of the placenta.
The number of prenatal visits and hospitalization days were significantly higher in comparison with the control group.
We are grateful for the opportunity to detail below the issues raised.
Major revisions
1) Study Design
I don’t get a clear sense of what you are trying to achieve with this study and the results and analyses you report don’t seem to have a clear value. To my mind your paper combines a descriptive report of outcomes from early-onset FGR and a case-control study. However, the statistical comparisons you make between your cases and controls don’t appear to answer any questions.
We aimed to find the prevalence of the condition in the unselected population delivered in out institution and to highlight the differences in terms of type of prenatal and postnatal care (in early-IUGR group versus normal cases). We revised the manuscript, to make this clearer.
In the Evidence Based Medicine paradigm, we approached the questions to help us grow our knowledge and practice.
We choose background questions (concerning general knowledge of early-IUGR). The question root was ”who, what, when, where, how, why” and the aspects of health were intended to be described in a specific setting (care of the disorder, test, treatment).
The design of the study didn’t target foreground questions, meaning specific knowledge questions that affect clinical decisions and include a broad range of biological, psychological, and sociological issues.
We used the PICO Model to help define the information need.
|
P |
Patient, Population, or Problem |
We described in detail a group of patients (similar population may be found) |
|
I |
Intervention, Prognostic Factor, or Exposure |
Main intervention (iatrogenic birth), prognostic factors (abnormal Doppler, hypertensive disorders), and exposure (smoking) were considered. |
|
C |
Comparison or Intervention |
The main alternative (continuation of pregnancy to term) is not suitable. The gold standard is not known. We compared prenatal features in two subgroups of affected patients within the study population. |
|
O |
Outcome (to measure or achieve) |
We hope to alter/improve the prenatal counselling of parents facing the same clinical condition. |
We confirmed that all ultrasound Doppler parameters showed abnormalities in the group investigated.
We edited table 1 and table 4 (which became table 3) and omitted table 2 in the revised form of the manuscript.
- Controls
You report the differences in Doppler velocimetry between cases and controls, but your cases were partially selected on the basis of Dopplers while your controls were selected not to have abnormal Dopplers, so they will by definition be different. Similarly, Table 1 reports co-morbidity for cases and controls, but you selected your controls to have no complications. You conclude from this that these co-morbidities are ‘risk factors’ but you cannot draw that conclusion from the study as designed.
We selected the study cases based exclusively on one fetal feature: the growth. We used Hadlock 4 nomogram to interpret it.
Indeed, in our case series of early-IUGR patients, all cases had abnormal Dopplers also. Yet, we wanted to confirm it - in the population studied, and to see if the degree of alteration might be related to the severity, to the outcome and/or to the gestational age at which the iatrogenic birth was required.
As work in progress, we currently perform a regression analysis to see which variables affect more the number of Neonatal Intensive Care Unit (NICU) days and the total number of hospitalization days.
Most probably, this will not help us for optimizing timing of delivery, but we hope this will provide additional information for future parents counselling, facing the same condition in the prenatal period.
Was the aim of the control group to provide a comparison when it comes to antenatal visits and newborn data? There may be some value in this for antenatal visits but has limited value for the newborn data.
We agree with the reviewer.
We accordingly edited the abstract and the text in the manuscript.
In our view, the new-born data (gestational age at delivery, birth weight, Apgar score, interventions in the neonatal period) may prove to be important and related in a direct manner - to the number of NICU days required and the total hospitalization days.
If you are going to have a control group, I would suggest reconsidering the group you have selected and providing more details about the selection. Why did you limit this to women with no complications having vaginal deliveries? Is this the most meaningful comparison? Were they selected at random or sequentially? Is there potential for bias in how they were selected? Do all women have second and third trimester ultrasound scans in your unit or were your controls selected because they had had scans? If so, why were they having scans?
We completely agree with the reviewer that choosing a completely normal (mothers, fetuses and normal mode of delivery) cohort may be seen as needless.
Yet, this was the design we choose, in order to highlight the practical consequences of this difficult to predict, difficult to diagnose and difficult to treat condition – early IUGR.
This is a rather biological failure, and - for the moment, as stated in the manuscript- we have no other treatment than closer prenatal follow-up and as appropriate as possible timing of delivery.
The normal cases were selected after the study was completed, sequentially. The first patient was selected at the study beginning (calendar’s date 1st Sept 2019). The last one – four months later. We had (as shown in the flow chart diagram) over 4000 deliveries in our unit in the study period. We suspect that we needed such a long period to be able to include in this control group the normal cases (56), due to the high incidence of C-section deliveries in our unit (almost 66%).
We do not deny the potential for bias in how they were selected, but we sincerely hope that it is limited: we selected consecutively, all cases that met the inclusion criteria (page 2, lines 95-99):
- completely followed-up and delivered in our hospital,
- healthy mothers,
- singleton pregnancy,
- normal fetuses (in terms of structure and growth curve),
- normal (vaginal) uncomplicated birth,
- term birth.
We added information regarding the control group in the manuscript.
Even if included in a low-risk pregnancy group at the first appointment, all women having prenatal care in our unit are offered and scheduled for the end-first-trimester (anomaly and ”genetic” scan), for a second trimester (structural survey - anomaly scan) and a third trimester (well-being) US scan. If the prenatal exams (dating and first trimester anomaly and genetic scan) offer completely normal data, for the second trimester scan the gestational age offered for scanning is 20-23 weeks, and for the third trimester – 29-33 weeks.
Indeed, if this data was not available, we did not include a specific case in the control group.
In the flow chart, we declared the no prenatal care group that delivered in our unit (130), the no FT dating group (210).
We presented this data, because we wanted to establish the prevalence of early-IUGR in our unselected population. We added this aim in the manuscript.
Unfortunately, we had cases lost to follow-up (18) and incomplete data (12) in the study group also.
- Cases
Was there a particular reason you limited your reported outcomes to those delivered before 32 weeks?
According to our team of neonatologists, it is customary to see 32 weeks as an important cut-off for preterm delivery. Furthermore, a restricted preterm baby (before 32 completed weeks) is considered extremely difficult to manage, and prone to major complications in the postnatal period and long-term sequala.
Therefore, we were interested in this particularly susceptible and delicate population of newborns. This is the reason for accepting to report data from a very small study.
When seeing these women during pregnancy we don’t necessarily know how things will progress and what will happen to the Dopplers, so providing information about all of these pregnancies could be helpful.
We thank the reviewer for considering our data.
Yes, indeed, the outcome of a particular pregnancy cannot be entirely predicted during the time interval of pregnancy evolving. As presented in the flow chart (page 7), we extracted from the final analysis a rather important number of fetuses (49), although they met the study inclusion criteria in the first three steps. The pregnancies evolving at and after 32 completed weeks were discarded (see also prestage I paragraph, page 3, lines 134-135 and page 5, lines 167-169). We felt that the remaining data are the most significant to be reported.
We could argue that in this cohort, many fetuses were either mild restricted, or constitutionally small, since all maintained well-being criteria and allowed continuing the pregnancy beyond 32 completed WG.
I think the value in your study comes from providing early-onset FGR outcome data from a middle-income country (the World Bank classify Romania as middle-income rather than developing). To this end it would be good to have more data about neonatal morbidity, even just short-term.
We thank the reviewer for the observation. We changed accordingly the title and the text in the manuscript.
In this short interval, we were able to augment the data on the new-borns also.
We added the data in the manuscript (see table 3 new-born data in pregnancies complicated with early IUGR – edited).
You compare Doppler measurements between those delivering 27-30 weeks and those delivering 30-32 weeks, but you also say third trimester scans were done at 28 weeks (or as soon as possible after enrolment). Does this mean some of the 27-30 week group don’t have an included measurement because they had delivered before 28 weeks? Where the measurements done at comparable gestations or were a lot of the 30-32 week group only enrolled at later gestations?
Yes. In this cohort we included exclusively babies born at 27 weeks and later.
By internal policy, we refer cases of severely early restricted fetuses needing iatrogenic delivery before 27 weeks.
These cases were also discarded.
In the study group, all US data were collected more than once, according to the study design. The statistical analysis was performed on the values at the beginning of the specific trimester (the second or the third).
Therefore, the processed values were the ones obtained at 22 weeks’ (21-23) or at the first prenatal visit in our unit - in all cases enrolled in the ST.
The second set of data entering in the final analysis were obtained at 28 weeks’ gestation respectively (in all cases already enrolled) or at the first prenatal visit (at enrolment) – in cases included (or referred to our unit) in the TT - after 28 completed weeks.
We revised the manuscript accordingly, to make clearer the study design.
Minor revisions
- Line 139: Did you include pregnancies with an EFW <10thand umbilical artery PI >95th in Stage 1? I know Francesc Figueras doesn’t include umbilical artery PI in his flow chart but he does include it in the table above.
We apologize, we don’t know what flow chart the reviewer is referring to. We cite the stage-based decision algorithm for management of FGR (figure 4) and the stage-based classification (table 2) in the reference 26. In this publication, in the text following the figure and the table, the authors state that in stage I – “either UtA, UA or MCA Doppler, or the CPR are abnormal”.
In our study, we defined the “prestage I” as estimated fetal weight (EFW) between 10th centile and the 3rd centile. And we defined stage I as EFW <3rd centile or CPR <5th centile or any UtA PI >95th centile.
- Figures 1-4: I’m not sure these are really necessary and your AEDV picture is not the clearest as there’s some background noise around the baseline.
We omitted all US figures.
- Lines 179-194: you don’t need to report this level of detail in your methods as it will become evident form your results what data you collected.
We omitted the details in this paragraph.
- Line 196: I don’t find your definition of GDM very helpful in this setting. The ADA paper you cite in turn cites the 4th International Workshop-conference on GDM from 1997 which in turn cites a number of other references and uses the phrase ‘varying degrees’ rather than ‘any degree’. I would argue all pregnant women are slightly more insulin resistant than they were before pregnancy because of the HPL. How did you test for it in your hospital? What cut-offs did you use?
In the revised form of our manuscript, we added the technique for routine screening for gestational diabetes mellitus (GDM) in our unit.
- Line 211: do you mean you tested for thrombophilias?
Yes, is this study we tested all patients for hereditary thrombophilia, regardless the personal history and parity.
- Line 225: I assume you mean p<0.05
Indeed, this was a typo. Thank you for noticing.
- Figure 5: I would say that assessing for eligibility and screening are the same thing. You might need to come up with different ways of talking about how and why you excluded people at different stages or put all the exclusions together.
We chose to use the flow chart diagram to represent (in a simple and easy to follow way) the entire work process. The diagram shows the flow of patients and how decisions affected the events surrounding them. In our view is depicting accurately each step in our work, and it provides a breakdown of the essential stages of the study.
Moreover, it helped us to reach the incidence of the condition in the unselected large population.
We apologize, but we sincerely cannot see how ”putting all the exclusions together” would help the report, or its illustrated representation.
We used the words ”assessed for eligibility” for the first step. At this point, we appraised the pregnancy files/documents and we estimated (by means of ultrasound) the fetal weight against the real gestational age (if previously documented). We also excluded a number of accurately dated pregnancies, subsequently diagnosed with congenital structural anomalies or chromosomal abnormalities at the 12-13 WG scan (page 3, lines 113-115).
In a subsequent stage (the second stage - ”participants screened”), we screened all fetuses falling under the 10th percentile by Hadlock nomogram. This means that we offered the entire US protocol. Moreover, we offered the complete work-up proposed in the study:
- we rescanned each case using the equipment destined to this study (page 3, lines 105-106)
- in order to provide homogenous US data for analysis the first author performed the scan or repeated the scan in referred cases (the senior consultant was involved in selected cases only)(page 3, lines 109-111)
- we performed in all cases Doppler interrogation, by the study design: both uterine arteries, umbilical artery, middle cerebral artery, and ductus venosus if appropriate; we calculated the cerebroplacental ratio (page 3, lines 119-122)
- we provided information to the attending physician
- we counselled parents, often in multidisciplinary teams
- we staged the case and filled-in the forms in the database (page 3, lines 129-145, page 4, page 5, lines 159-164)
- we scheduled the next prenatal visit/visits or we admitted the case for hospitalization
- we collected additional data (demographic and baseline)
- we scheduled, performed and documented OGTT
- we measured as described the maternal blood pressure (page 6, lines 198-200)
- we performed laboratory tests for hereditary thrombophilia
Unfortunately, after completing this stage we lost some cases also, due to infections diagnosed (mainly chorioamnionitis) and due to (minor and major) structural malformations diagnosed in the second trimester.
We ended-up in the third stage with ”patients considered for the study”. From this group, we excluded 33 cases (lost to follow-up, incomplete data due to human error and withdrawal of the informed consent).
In the final stage we tried to process data from the remaining 85 cases. Eventually, we decided to keep for final analysis the data on patients requiring iatrogenic birth before 32 WG only, as we explained above.
We apologize, but we are unable to modify figure 5. If the reviewer has any suggestions, we are open to improve it.
- Line 259: ‘significand’
Indeed, this was another typo. Thank you. We omitted this phrase for the above explained reason.
- Figure 6: how are you defining placental and non-placental? It isn’t completely clear what you’re comparing in these figures but the lines between the medians are not appropriate, as this would be for changes in one group over time. I prefer box plots that are overlayed with the individual data points, as they give a clearer sense of the spread of the data.
At the beginning of this study, we wanted to find out whether placental laterality and ipsilateral isolated abnormal uterine artery Doppler (waveform and PI) can be used as a predictor for the early-IUGR occurrence. Thus, we added to the US scan form (completed by the operator for this study) this information: the placenta (lateral) localization.
The observer diagnosed ”lateral placenta” if more than half of the placenta was seen on US on one side of the uterine cavity only. Among the 36 reported cases, we found in 30 cases exclusively laterally located placentas. The corresponding (right or left) uterine artery was named ”placental”. The other one was named ”non-placental” uterine artery on the US form.
In the 6 remaining cases (with rather central - anterior, posterior or fundal- placenta) the operator decided on subjective criteria the assignment of the uterine arteries.
We edited the text in the manuscript, to clarify this issue.
We completely agree that a graphic representation over time would have been superior.
As we described in the method section of the manuscript, we used for the final analysis one set of data only, for each trimester: the second and the third.
Round 2
Reviewer 2 Report
Thank you for considering our comments and suggested changes.
I have a few minor comments and have included some suggested edits to the text on the pdf. I can't seem to attach it but will try to find another way to upload it.
1) Lines 52 and 53: you say GA and USS findings are independent predictors of adverse outcome, but I cannot see any statistical analysis demonstrating this. Your analyses involve comparing two groups and showing differences in the parameters (e.g. comparing length of stay for 27-30 vs 30-32) but to determine predictors and to work out whether they were independent predictors you would need to use logistic or linear regression. I don't think you necessarily need to add these analyses, you could just rephrase this sentence.
2) Line 27 etc.: you say you have aimed to determine the prevalence, but I can't see any evidence of this. When it comes to pregnancy, I would generally talk about incidence rather than prevalence, given the finite nature of pregnancy. Either way you would need to provide data on the population. Does n=4011 total deliveries include all of the women pregnant during this interval? Do you receive referrals from other lower level units? (in which case the total population you are drawing from would be larger) Could there have been any 'missed' cases e.g. stillbirths with undiagnosed IUGR? Again, you don't necessarily need to add these, just consider the terms you use.
3) Line 94: you say you used the Hadlock 4 formula for estimating fetal weight, but which chart did you use for determining EFW centile? If you use the Hadlock chart, the reference for this is the 1991 paper.
4) Line 232: when you say you 'considered' hereditary thrombophilia do you mean you tested for it?
5) Line 261: I'd suggest using 'neonatal infection' rather than 'neonatal infection or sepsis' because the latter raises the question 'which ones had sepsis?'
6) Table 3/ Line 333: I'd use capitals for all of the complications
7) Figures 2-4: provide p values for the statistical tests. You say in line 383 that there were no statistically significant differences between groups, but several of your box plots have asterisks on them, which are usually used to indicate a statistically significant difference. If these asterisks are there in error and there genuinely were no statistically significant differences you need to consider rephrasing. For example, you say there were "much more visits in the very early suspected IUGR subgroup" but if there is no statistically significant difference you would be be better rephrasing this as "Although the median number of prenatal visits was higher in very early suspected IUGR than early suspected IUGR, this difference did not reach statistical significance (10 vs 6, p=??)"
8) Discussion: this could do with cutting down and restructuring to make it clearer. One way to structure a discussion is (1) main findings (very brief, not just repeating your whole results section) (2) how do our findings compare to the published literature? If there are differences why might that be? (3) strengths and limitations (4) what do our results mean in practice? What impact might they have? (5) future work. Don't bring in any other questions or areas of the literature that don't directly relate to what you have done - this should either be in the introduction or should be left out (however interesting it may be!) I'd suggest at least halving it.

Author Response
Thank you for considering our comments and suggested changes.
I have a few minor comments and have included some suggested edits to the text on the pdf. I can't seem to attach it but will try to find another way to upload it.
We were able to see the suggested edits un the pdf document. We revised the manuscript accordingly.
1)Lines 52 and 53: you say GA and USS findings are independent predictors of adverse outcome, but I cannot see any statistical analysis demonstrating this. Your analyses involve comparing two groups and showing differences in the parameters (e.g. comparing length of stay for 27-30 vs 30-32) but to determine predictors and to work out whether they were independent predictors you would need to use logistic or linear regression. I don't think you necessarily need to add these analyses, you could just rephrase this sentence.
We rephrased it.
2) Line 27 etc.: you say you have aimed to determine the prevalence, but I can't see any evidence of this. When it comes to pregnancy, I would generally talk about incidence rather than prevalence, given the finite nature of pregnancy. Either way you would need to provide data on the population. Does n=4011 total deliveries include all of the women pregnant during this interval? Do you receive referrals from other lower level units? (in which case the total population you are drawing from would be larger) Could there have been any 'missed' cases e.g. stillbirths with undiagnosed IUGR? Again, you don't necessarily need to add these, just consider the terms you use.
We replaced the word ”prevalence”. We edited the workflow chart, to make it clearer.
No, we don’t have data on lower-level units’ deliveries. We cannot exclude missed cases (undiagnosed IUGR resulting in stillbirths), due to incomplete coverage (the significant population of pregnant women in rural areas, having no/suboptimal prenatal care). As supplemental information: in hospitals and units ranking lower, it is mandatory to refer cases with increased risk for birth before 32 weeks and below 2000 g - estimated fetal weight.
All referred to us cases (regardless the gestational age at recruitment) were subsequently managed and delivered in our hospital, with very few exceptions, explained in the manuscript: 3 intrauterine deaths (among them – two cases of fetal abandonment) and two referrals for extreme severity, requiring delivery before 27 weeks. All these cases were extracted from the final analysis.
3) Line 94: you say you used the Hadlock 4 formula for estimating fetal weight, but which chart did you use for determining EFW centile? If you use the Hadlock chart, the reference for this is the 1991 paper.
We used the Hadlock chart, included as a default setting (the US system used for the study). We replaced the reference 12.
4) Line 232: when you say you 'considered' hereditary thrombophilia do you mean you tested for it?
We rephrased the sentence in the method.
As we explained in the first round of the review, we screened all fetuses falling under the 10th centile by Hadlock nomogram for hereditary thrombophilia, in the second stage of the study (”participants screened” – 179, shown in the flow chart), regardless the personal history and parity.
5) Line 261: I'd suggest using 'neonatal infection' rather than 'neonatal infection or sepsis' because the latter raises the question 'which ones had sepsis?'
We omitted the word ”sepsis”.
6) Table 3/ Line 333: I'd use capitals for all of the complications
We made the required change.
7) Figures 2-4: provide p values for the statistical tests. You say in line 383 that there were no statistically significant differences between groups, but several of your box plots have asterisks on them, which are usually used to indicate a statistically significant difference. If these asterisks are there in error and there genuinely were no statistically significant differences, you need to consider rephrasing. For example, you say there were "much more visits in the very early suspected IUGR subgroup" but if there is no statistically significant difference you would be better rephrasing this as "Although the median number of prenatal visits was higher in very early suspected IUGR than early suspected IUGR, this difference did not reach statistical significance (10 vs 6, p=??)"
Figures 2-4 display the distribution of data based on “minimum” value, first quartile median, third quartile and “maximum” value. Based on Kruskal-Wallis Test the differences between medians are statistically significant (p<0.01).
We apologise for the mistake; we changed the wording in the revised form of the manuscript.
Following the text: ”In the extremely early suspected IUGR group we found a higher median than in the early suspected IUGR. The boxplot of Umbilical Artery percentile revealed that in the very early suspected IUGR group we found a higher median than in the early IUGR group” (lines 297 – 300 in the revised form), we added: ”Based on Kruskal-Wallis Test the differences between medians are statistically significant (p<0.01)”.
Following the text: ”Total prenatal visits in pregnancy and of TT visits revealed an increased number of medical visits in the very early suspected IUGR subgroup (Figure 3)” (lines 307 – 308 in the revised form), we added: ”Based on Kruskal-Wallis Test the differences between medians are statistically significant (p<0.01)”.
Following the text: ”In the very early suspected IUGR subgroup we had higher median values for NICU and total hospitalization days (21.5 days and 49 days respectively) compared to the early suspected IUGR subgroup. (Figure 4)” (lines 312 – 314 in the revised form), we added: ”Based on Kruskal-Wallis Test the differences between medians are statistically significant (p<0.01)”.
We add the following data:
WORKSHEET 1
Kruskal-Wallis Test: TT non-placental UtA - MoM versus Category
Descriptive Statistics
|
Category |
N |
Median |
Mean Rank |
Z-Value |
|
GA 27-30 |
12 |
98.5 |
23.8 |
2.11 |
|
GA 30-32 |
24 |
96.5 |
15.9 |
-2.11 |
|
Overall |
36 |
|
18.5 |
|
Test
|
Null hypothesis |
H₀: All medians are equal |
||||
|
Alternative hypothesis |
H₁: At least one median is different |
||||
|
Method |
DF |
H-Value |
P-Value |
|
|
|
Not adjusted for ties |
1 |
4.47 |
0.01 |
|
|
|
Adjusted for ties |
1 |
4.93 |
0.01 |
|
|
WORKSHEET 1
Kruskal-Wallis Test: TT placental UtA - MoM versus Category
Descriptive Statistics
|
Category |
N |
Median |
Mean Rank |
Z-Value |
|
GA 27-30 |
12 |
99.5 |
19.3 |
0.30 |
|
GA 30-32 |
24 |
99.0 |
18.1 |
-0.30 |
|
Overall |
36 |
|
18.5 |
|
Test
|
Null hypothesis |
H₀: All medians are equal |
||||
|
Alternative hypothesis |
H₁: At least one median is different |
||||
|
Method |
DF |
H-Value |
P-Value |
|
|
|
Not adjusted for ties |
1 |
0.09 |
0.1 |
|
|
|
Adjusted for ties |
1 |
0.10 |
0.1 |
|
|
WORKSHEET 1
Kruskal-Wallis Test: Prenatal visits in TT versus Category
Descriptive Statistics
|
Category |
N |
Median |
Mean Rank |
Z-Value |
|
GA 27-30 |
12 |
10 |
18.5 |
0.00 |
|
GA 30-32 |
24 |
6 |
18.5 |
0.00 |
|
Overall |
36 |
|
18.5 |
|
Test
|
Null hypothesis |
H₀: All medians are equal |
||||
|
Alternative hypothesis |
H₁: At least one median is different |
||||
|
Method |
DF |
H-Value |
P-Value |
|
|
|
Not adjusted for ties |
1 |
0.00 |
0.01 |
|
|
|
Adjusted for ties |
1 |
0.00 |
0.01 |
|
|
WORKSHEET 1
Kruskal-Wallis Test: NICU versus Category
Descriptive Statistics
|
Category |
N |
Median |
Mean Rank |
Z-Value |
|
GA 27-30 |
12 |
1.5 |
11.8 |
-2.72 |
|
GA 30-32 |
24 |
13.5 |
21.9 |
2.72 |
|
Overall |
36 |
|
18.5 |
|
Test
|
Null hypothesis |
H₀: All medians are equal |
||||
|
Alternative hypothesis |
H₁: At least one median is different |
||||
|
Method |
DF |
H-Value |
P-Value |
|
|
|
Not adjusted for ties |
1 |
7.39 |
0.001 |
|
|
|
Adjusted for ties |
1 |
7.57 |
0.001 |
|
|
WORKSHEET 1
Kruskal-Wallis Test: Hospitalisation versus Category
Descriptive Statistics
|
Category |
N |
Median |
Mean Rank |
Z-Value |
|
GA 27-30 |
12 |
6.5 |
11.4 |
-2.87 |
|
GA 30-32 |
24 |
49.0 |
22.1 |
2.87 |
|
Overall |
36 |
|
18.5 |
|
Test
|
Null hypothesis |
H₀: All medians are equal |
||||
|
Alternative hypothesis |
H₁: At least one median is different |
||||
|
Method |
DF |
H-Value |
P-Value |
|
|
|
Not adjusted for ties |
1 |
8.23 |
0.001 |
|
|
|
Adjusted for ties |
1 |
8.37 |
0.001 |
|
|
Kruskal-Wallis Test: Total prenatal visits versus Category
Descriptive Statistics
|
Category |
N |
Median |
Mean Rank |
Z-Value |
|
GA 27-30 |
12 |
18.0 |
23.8 |
2.11 |
|
GA 30-32 |
24 |
11.5 |
15.9 |
-2.11 |
|
Overall |
36 |
|
18.5 |
|
Test
|
Null hypothesis |
H₀: All medians are equal |
||||
|
Alternative hypothesis |
H₁: At least one median is different |
||||
|
Method |
DF |
H-Value |
P-Value |
|
|
|
Not adjusted for ties |
1 |
4.47 |
0.01 |
|
|
|
Adjusted for ties |
1 |
4.64 |
0.001 |
|
|
8) Discussion: this could do with cutting down and restructuring to make it clearer. One way to structure a discussion is (1) main findings (very brief, not just repeating your whole results section) (2) how do our findings compare to the published literature? If there are differences why might that be? (3) strengths and limitations (4) what do our results mean in practice? What impact might they have? (5) future work. Don't bring in any other questions or areas of the literature that don't directly relate to what you have done - this should either be in the introduction or should be left out (however interesting it may be!) I'd suggest at least halving it.
We revised the discussion section. We reduced the number of words (2130 to 1180).
